# Experimental evidence demonstrating how freeze-thaw patterns affect spoilage of perishable cached food

**Karen Ong** *, **D. Ryan Norris**

Department of Integrative Biology, University of Guelph, Guelph, Ontario, Canada

* ongk@uoguelph.ca

## Abstract

For the small number of temperate and boreal species that cache perishable food, previous research suggests that increasing freeze-thaw events can have a negative impact on fitness by degrading the quality of cached food. However, there is no experimental evidence that directly links freeze-thaw events to cache quality. To examine how the timing, frequency, duration, and intensity of freeze-thaw events influenced cached food mass loss, a proxy for caloric content, we conducted a series of month-long laboratory experiments by placing simulated caches (raw chicken placed between two pieces of black spruce *Picea mariana* bark) in programmable freezers. Freeze-thaw treatments were modelled after weather data from Algonquin Provincial Park, Ontario, where a population of Canada jays (*Perisoreus canadensis*), a species that caches perishable food for overwinter survival and to support late-winter breeding, has declined by > 70% since the 1980s. First, we found no evidence that an increased frequency of freeze-thaw events influenced mass loss, suggesting that microstructural damage caused by crystal reformation does not significantly influence cache quality. Instead, our experimental results demonstrated that mass loss was positively influenced by longer individual thaws, which likely reflects increased microbial growth, oxidation, and progressive drip loss. We also found that caches lost more weight when subjected to early freeze-thaw events compared to late freeze-thaw events. Finally, we show that milder freezes led to less mass loss and, unexpectedly, warmer than average thaws post-freeze also led to less mass loss. Our results suggest that longer thaw periods post-freezing and milder freezes cause or lead to significantly increased spoilage of perishable cached food. All of these temperature-related conditions are closely associated with long-term changes in climate and, thus, the effects on cache degradation reported in these experiments should be applicable to species caching perishable food in the wild.

## Introduction

Food caching, the storage of food for later consumption, is used predominantly by mammals and birds to overcome periods of food scarcity [1,2]. Whether species store food over the short-term (hours to days) or the long-term (weeks to months), their caches are exposed

**Data availability statement:** All data are contained within the manuscript and supporting information files.

**Funding:** The author(s) received no specific funding for this work.

**Competing interests:** The authors have declared that there are no competing interests.

to environmental conditions that result in progressive deterioration through spoilage [1,2]. Spoilage is defined as the change in a food's sensory attributes (i.e., flavour, odour, colour) that results from physical and biochemical processes such as damage to the cellular membrane, autooxidation, and microbial proliferation [3,4]. The spoilage rate of food is directly related to the free water available for biochemical reactions which is quantified as water activity, $a_w$ [3,5]. Food with a lower $a_w$ can remain stable for longer periods (typically called 'non-perishables'; e.g., seeds, nuts, dried vegetation) compared to food with a higher free water content ('perishables'; e.g., fruits, meat). Food-caching species employ a variety of solutions to reduce the $a_w$ of their caches. For example, American pikas (*Ochontona princeps*) and red squirrels (*Tamiasciurus hudsonicus*) dry vegetation and fungi on rocks and bare tree branches, respectively [6–8]. Other species, like the Florida scrub-jay (*Aphelocoma coerulescens*), favour drier or shaded (cooler) microhabitats for food storage that slow the spoilage of their cache compared to hot and humid conditions [3,9–11]. Despite these innovative species-specific strategies, shifts in temperature and precipitation patterns associated with climate change are creating additional challenges for the prevention of cache spoilage.

Climate change is expected to affect food-storing species, especially those that store perishable foods, because unstable foods are sensitive to increased ambient temperatures and fluctuating weather conditions. Range retractions in wolverines (*Gulo gulo L.*) [12], for example, have been attributed to decreased spring snow cover that is critical for proper insulation of overwintering food stores [13–15]. Stable subfreezing temperatures prevent freeze-thawing which destroys the tissues in food, releasing precursor compounds that expedite several pathways of spoilage. As food freezes, ice crystals form within protein and connective tissue, initiating denaturation while also setting the stage for accelerated secondary lipid oxidation post-thaw through the release of pro-oxidants [16,17], altering the bioavailability and digestibility of amino acid residues and proteins [18]. For meat, the destruction of these proteins decreases its water holding capacity, which leads to increased drip loss when the meat is thawed [19,20]. For example, frozen-thawed ostrich meat has a lower pH and water holding capacity but higher drip loss than its fresh counterpart [21]. Similar results have been found in frozen-thawed versus fresh pork meat [22]. The exudate that drips from the damaged cells contains valuable water, vitamins, minerals, and amino acids [17,23,24]. Along with the loss of nutrients, the holes left from ice crystals melting provide access by microbes. Unlike fungi, bacteria are incapable of invasive action into tissues, making them dependent on extrinsic forces, such as freeze-thaw events, to mechanically disrupt tissue [24]. Early fall freeze-thaw events have already been associated with reduced survival of pygmy owls (*Glaucidium passerinum*) and the mechanistic link in this relationship is suspected to be a decrease in cache quality needed for overwinter survival [25].

The Canada jay (*Perisoreus canadensis*) is another long-term perishable food-caching species experiencing population decline [26,27]. This passerine is a year-round resident of boreal and subalpine forests in North America that stores vertebrate flesh, berries, arthropods, and fungi for overwinter survival and late-winter breeding [28–30]. At the southern edge of their range in central Ontario, a population of Canada jays in Algonquin Provincial Park (45°84'N, 78°38'W; hereafter 'Algonquin Park') has declined by over 70% in the last three decades [27]. To explain this decline, Waite & Strickland [28] presented the '*hoard-rot hypothesis*', which proposes that warmer fall temperatures were exacerbating spoilage of perishable food caches that Canada jays depend on for rearing young in their late-winter nesting season. Sutton et al. [26,27] later showed a negative correlation between the number of fall and winter freeze-thaw events and late-winter reproductive performance, suggesting that conditions during the fall influence the quality or quantity of cached food that is used for reproduction 5–7 months later. Two food supplementation experiments have provided evidence that the late-winter

reproduction of Canada jays is limited by food availability [31,32]. An overall warming trend associated with climate change in Algonquin Park and the declining Canada jay population provide indirect evidence for the negative effects of freeze-thaw events on fitness and population abundance [27,33]. However, there is no direct evidence linking freeze-thaw events to the quality of cached food in Canada jays or any other food-caching species. Moreover, previous studies on food for human consumption suggest that there may be multiple ways in which freeze-thaw events act to degrade cached food that is independent of simply the frequency at which they occur [34–39].

In this study, our goal was to examine whether the timing, frequency, thaw duration, and intensity of freeze-thaw events affected cache quality. In doing so, we experimentally examined the following hypotheses:

(*H1*) The '*exacerbation hypothesis*' proposes that the presence of an early freeze-thaw event, such as one associated with an early-season cold spell, increases the total duration of thaw post-freeze, thereby increasing spoilage rates compared to a later freeze-thaw event. Spoilage rates are higher post-freeze because freezing inflicts microstructural damage to the food that facilitates microbial spoilage and accelerates oxidation. An earlier freeze-thaw event simply initiates the process of accelerated spoilage sooner, reducing cache quality earlier than a late freeze-thaw.

(*H2*) The '*frequency hypothesis*' proposes that, up to a point, subsequent freezing events inflict additional microstructural damage, leading to greater post-freezing spoilage rates with each subsequent event. Repeated freeze-thaw cycles are associated with larger ice crystal diameter following recrystallization, a greater volume of ice, and increased porosity of the food item causing greater structural damage that facilitates other processes that decrease food quality [40].

(*H3*) The '*continuous thaw hypothesis*' proposes that for the same total thaw hours post-freeze, longer individual thaw periods allow spoilage to increase whereas short bursts of thawing temperatures do not allow the metabolic activity of spoilage microbes to increase to impactful levels.

(*H4*) The '*thaw intensity hypothesis*' proposes that a more intense thaw subjects food to higher above-freezing temperatures which are better tolerated by microbes, resulting in higher rates of microbial growth and lipid oxidation [41–43].

(*H5*) The '*freeze intensity hypothesis*' proposes that a faster decrease in temperature, or more intense freeze, results in a colder minimum temperature which would create smaller ice crystals that are more uniformly distributed inside and outside the cell, inflicting less physical damage to the tissue, thereby minimizing drip loss and other characteristics associated with food quality [44–46]. Additionally, a more intense freeze can kill, injure, and arrest microbes, reducing the viable microbial population for growth when thawing temperatures are restored [47–49].

We tested predictions from each of these hypotheses by conducting a series of month-long simulated freeze-thaw experiments on commercial raw chicken in programmable freezers using weight as a proxy for cache quality [50].

## Methods

We conducted a series of freeze-thaw simulations to examine the effect of freeze-thaw events on cache quality using chest freezers programmed by Titan 900 (v 9.0.12.2; Argus System

Controls, Winnipeg, Manitoba, Canada) at the University of Guelph (Guelph, Ontario, Canada). To test all predictions (see Introduction) with ecological validity, laboratory simulations were based on values of maximum temperatures, minimum temperatures, and durations of freeze and thaw phases extracted from hourly climate data of the Environment Canada Algonquin Park East Gate weather station, Algonquin Park, Ontario (45°32'N, 78°54'W; https://weather.gc.ca/city/pages/on-29_metric_e.html) between 2004 – 2019. The freezing point we used to separate freeze and thaw phases was −1.9 °C because this is the lower range for the freezing point of meat [51]. From weather station data, we initially extracted freeze-thaw events from October, when caching behaviour of Canada jays is prominent, to December, the month of fall-to-winter transition. Using a one-way ANOVA, we found that month was a significant predictor of freeze-thaw events ($F_{2, 45}$ = 18.17, $p$ < 0.0001). A post hoc Tukey test revealed that November had significantly more freeze-thaw events than both October ($p$ < 0.0001) and December ($p$ = 0.0002). As a result, freeze-thaw simulations were modelled after the month of November because this is the month that likely accounts for much or most of the spoilage in caches (Tables S1, S2).

In nature, a single freeze-thaw event consists of a period of sub-freezing temperatures (the 'freeze phase') followed by a period of above-freezing temperatures (the 'thaw phase'). To simplify our laboratory experiments, we defined average freeze and thaw temperatures using the mean minimum and mean maximum temperatures of the observed minima and maxima for each freeze and thaw event in Algonquin Park between 2004–2019 (Table S2). Three separate 720-hr long experiments were designed to examine whether the timing (experiment 1; *H1 'exacerbation hypothesis'*), number (experiment 2; *H2 'frequency hypothesis'*), duration (experiment 2; *H3 'continuous thaw hypothesis'*), and the intensity (experiment 3; *H4 'thaw intensity hypothesis'* and *H5 'freeze intensity hypothesis'*) of freeze-thaw events influenced the quality of simulated caches (Fig 1).

For all experiments described below, we simulated Canada jay caches by placing a piece of organic raw chicken that weighed 1.20 g (Rowe Farms, Guelph, Ontario, Canada) between two pieces of black spruce (*Picea mariana*) bark collected from the Guelph Arboretum (43°54'N, 80°21'W). The combined ranges of black and white (*P. glauca*) spruce define the range of the boreal morphotype of the Canada jay [52,53] and have been experimentally shown to be the most effective of nine tree species sampled in Algonquin Park at preserving caches of perishable food [54]. The black spruce bark presumably contributed environmental microbes and volatile resins to the chicken, creating a more realistic simulated cache [55]. Nine replicates were included in each treatment and the weight of each replicate was taken to the nearest 0.01g ± 0.03 immediately before and after the experiment was finished using a Fuzion digital pocket scale (model TU-200-RED, 2021). Proportional weight loss was calculated by modifying the equation to calculate drip loss (proportional weight loss = weight lost [g]/starting weight [g]) to assess changes in cache quality. Drip loss (drip loss (%) = proportional weight loss x 100%; [56,57] is directly related to meat pH and oxidation [21,58].

To examine *H1*, the *'exacerbation hypothesis'*, we implemented two treatments: an 'early freeze-thaw' treatment where eight consecutive freeze-thaw events (41-hr freeze; 16-hr thaw; last thaw was 209 hrs) began at the 6th hr of the 720-hr period (337 hrs of post-freeze thaw conditions) and a 'late freeze-thaw' treatment where eight freeze-thaw events occurred at the 211th hr (132 hrs of post-freeze thaw conditions; Fig 1A, Table S3).

To examine *H2*, the *'frequency hypothesis'* and *H3*, the *'continuous thaw hypothesis'*, we conducted three treatments that started with a freeze and had an identical total duration below freezing (368 hrs total) and above-freezing (352 hrs total) post-freeze (Table S4). The first treatment level was the 'low frequency freeze-thaw' that had 8 freeze-thaw events (46.5-hr freeze; 44-hr thaw), a 'medium frequency freeze-thaw' that had 15 freeze-thaw events (24.5-hr

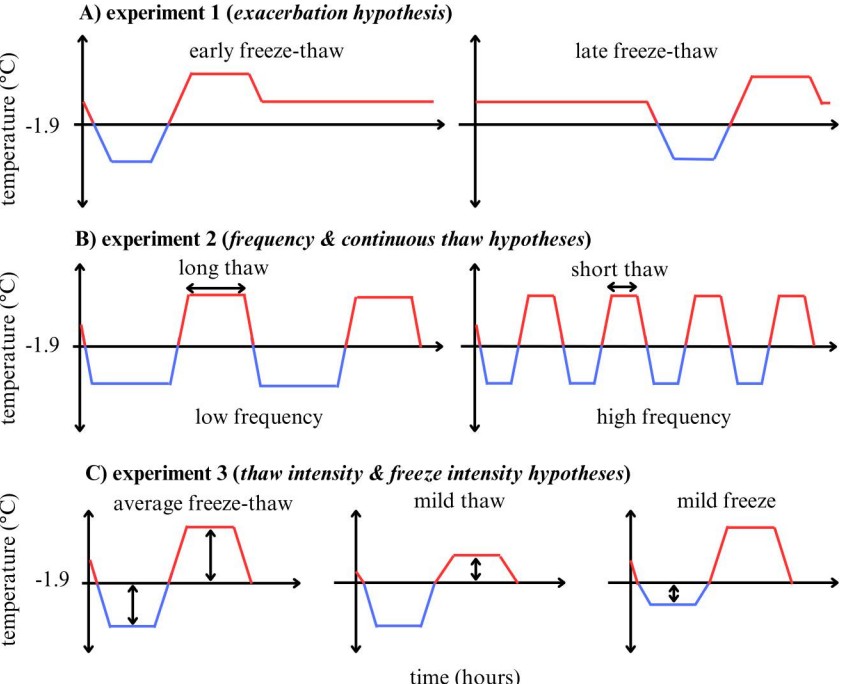

**Fig 1. Schematic diagrams of freeze-thaw manipulations conducted in a programmable freezer.** Blue lines denote freeze phases and red lines denote thaw phases with −1.9 °C as the initial freezing point (lower freezing point of meats). (A) Experiment 1 examined the '*exacerbation hypothesis*': held at a constant temperature introducing the first freeze-thaw event at the beginning or part way through the experimental period. (B) Experiment 2 examined the '*frequency hypothesis*' and '*continuous thaw hypothesis*': the number of freeze-thaw events was increased by shortening the duration of individual thaw phases (indicated by the black horizontal arrows) to maintain a consistent total number of hours above and below freezing across treatments. (C) Experiment 3 examined the '*thaw intensity hypothesis*' and the '*freeze intensity hypothesis*': the 'mild thaw' treatment had the maximum temperature lowered and the 'mild freeze' treatment had the minimum temperature raised by halving the freezing and thawing rates from 0.8 °C/hr to 0.4 °C/hr. The black vertical arrows represent the amplitude that was manipulated.

freeze; 24-hr thaw; Table S2), and a 'high frequency freeze-thaw' (16.5-hr freeze, 16-hr thaw). All freeze-thaw events reached a maximum of 4.5 °C and minimum of −7.5 °C (Fig 1B). *H2* would be supported if greater weight loss was seen in the high frequency freeze-thaw treatment, indicating repeated microstructural damage significantly decreases cache quality. *H3* would be supported if greater weight loss was seen in the low frequency freeze-thaw treatment, indicating that longer durations of higher spoilage rates post-freeze would be more detrimental to cache quality than repeated microstructural damage.

To examine *H4*, the '*thaw intensity hypothesis*', and *H5*, the '*freeze intensity hypothesis*', we manipulated the intensity of freeze-thaw events to create three treatments, all of which consisted of eight 73-hr freezes and eight 16-hr thaws across treatments: an 'average freeze-thaw' with a minimum temperature of −6.7 °C and maximum temperature of 4.5 °C (thawing and freezing rate of 0.8 °C/hr), a 'mild freeze' with a minimum temperature of −4.3 °C (freezing rate of 0.4 °C/hr) and maximum of 4.5 °C (thawing rate of 0.8 °C/hr), and a 'mild thaw' treatment with a minimum temperature of −6.7 °C (freezing rate of 0.8 °C/hr) and a maximum temperature of 1.3 °C (thawing rate of 0.4 °C/hr; Fig 1C, Table S5). *H4* would be supported if the average freeze-thaw treatment (max. temp.: 4.5 °C) resulted in greater weight loss than the mild thaw treatment (max. temp.: 1.3 °C) given that freeze conditions were the same between these two treatments (min. temp.: −6.7 °C in both). *H5* would be supported if the mild freeze

treatment (min. temp.: −4.3 °C) resulted in greater weight loss than the average freeze-thaw treatment (min. temp.: −6.7 °C) given that thaw conditions were the same between these two (max. temp.: 4.5 °C in both).

For all experiments, we also included controls that lacked freeze-thaw events for the entire 720 hrs. Experiment 1 and 3 included a high temperature control where temperature was held at 1.1 °C (3 °C above the average freezing point of meat) for the duration of the 720-hr period and a low temperature control where temperature was held at −4.9 °C (3 °C below the average freezing point of meat). Experiment 2 included a mild freeze control held at −1.6 °C, the mean temperature of all three freeze-thaw treatments. We chose to use different controls in Experiment 2 to experimentally test the effect of freeze-thaw events against average ambient temperatures, because Sutton et al. [27] found support that they may contribute more to cache spoilage.

Processes such as oxidation and microbial spoilage are highly dependent on free water available (unbonded). A high $a_w$ would indicate that favourable conditions were provided by the caches for microbial growth, which would not be measured with weight loss, our main metric for spoilage. In attempt to address this, we used an AquaLab Vapor Sorption Analyzer (Decagon Devices, Inc., Pullman, WA, USA) provided by the University of Guelph Food Science department. However, the accuracy of the equipment decreases significantly once the $a_w$ is below 0.8 so we could not rely on the readings that ranged between 0.6 to 0.7. As the $a_w$ could not be reliably measured, we relied on the fresh weights only. We chose not to use freeze-dried weights because we were interested in the state of the cache prior to it being consumed.

## Statistical analyses

To analyze how different freeze-thaw treatments affected cache weight loss, we used generalized linear models (GLMs) implemented in the stats package in R Studio 4.3.1 [59]. In all models, proportional weight loss was the response variable (quasibinomial distribution) with treatment as the predictor variable. After models were run, the β coefficient was calculated using beta function from R's reghelper package [60] to quantify the effect size of each level of a given treatment on proportional weight loss relative to a base treatment level. Following this, we performed pairwise comparisons of the estimated marginal means (EMMs) at the treatment level using the emmeans R package with a 95% confidence interval and p-values that were adjusted using the Tukey method [61].

## Results

### Experiment 1: Exacerbation hypothesis (*H1*)

Consistent with predictions from *H1*, caches subjected to the early freeze-thaw treatment lost significantly more weight than caches in the late freeze-thaw treatment (β = 0.11 ± 0.02, *p* < 0.0001) and also lost more weight than caches in both controls (Fig 2A, Tables 1 and S6). The late freeze-thaw caches did not differ significantly in weight loss from either the low (*z*(inf) = 0.5, *p* = 0.9; Fig 2A) or high (*z*(inf) = −1.5, *p* = 0.5; Fig 2A) temperature controls.

### Experiment 2: Frequency (*H2*) & continuous thaw (*H3*) hypotheses

Contrary to predictions from *H2* but consistent with predictions from *H3*, there was greater weight loss in the low frequency freeze-thaw treatment compared to both the medium and high frequency freeze-thaw treatments (Tables 2 and S7). Weight loss of caches in the high frequency freeze-thaw treatment did not differ significantly from weight loss in the medium

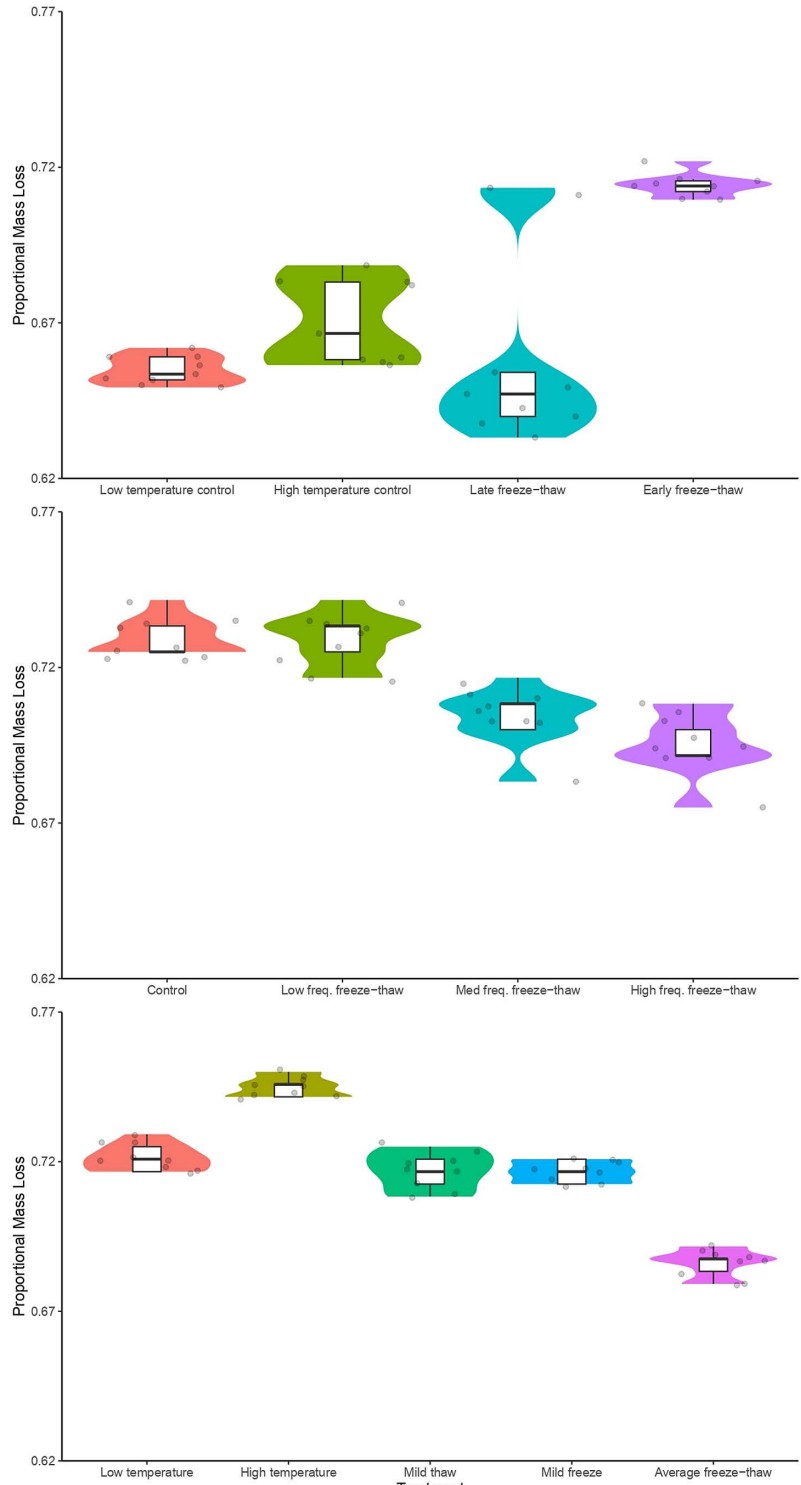

**Fig 2. Proportional mass loss of organic raw chicken breast with a starting mass of 1.20 grams.** (A) Simulated freeze-thaw experiment 1, examining the '*exacerbation hypothesis*'. Mean proportional mass loss was as follows: "Low temperature" (−5 °C, no freeze-thaw events) = 0.65 ± 0.005, "High temperature" (1 °C, no freeze-thaw events) = 0.67 ± 0.01, "Late freeze-thaw" = 0.66 ± 0.03, and "Early freeze-thaw" = 0.71 ± 0.004. (B) Simulated freeze-thaw experiment 2, examining the '*frequency hypothesis*' and the '*continuous thaw hypothesis*'. Mean proportional mass loss was as follows: "Control" (−1.62 °C, no freeze-thaw events) = 0.73 ± 0.006, "Low frequency freeze-thaw" (8 freeze-thaw events, 44-hour thaws) = 0.73 ± 0.008, "Med frequency freeze-thaw" (15 freeze-thaw events; 24-hour thaws) = 0.70 ±

0.009, "High frequency freeze-thaw" (22 freeze-thaw events; 16-hour thaws) = 0.70 ± 0.01. (C) Simulated freeze-thaw experiment 3 examining the *'thaw intensity hypothesis'* and *'freeze intensity hypothesis'*. Mean proportional mass loss was as follows: "Low temperature" (−4.9 °C, no freeze-thaw events) = 0.72 ± 0.004, "High temp" (1.1 °C, no freeze-thaw events) = 0.74 ± 0.003, "Mild thaw" (freezing rate of 0.8 °C/hr, thawing rate of 0.4 °C/hr) = 0.72 ± 0.006, "Mild freeze" (freezing rate of 0.4 °C/hr, thawing rate of 0.8 °C/hr) = 0.72 ± 0.004, and "Average freeze-thaw" (freezing and thawing rate of 0.8 °C/hr) = 0.69 ± 0.005.

**Table 1. Generalized linear model output of freeze-thaw treatments on cache weight loss to examine the '*exacerbation hypothesis*'.** Treatment was used as the predictor variable and proportional weight loss was used as the dependent variable with a quasibinomial distribution.

| Fixed effect | β Estimate | Std. Error | t-value | *p*-value |
|---|---|---|---|---|
| Intercept (late freeze-thaw) | 0.731 | 0.012918 | 56.59 | < 0.0001 |
| Early freeze-thaw | 0.114 | 0.016219 | 7.00 | < 0.0001 |
| Low temp. control | −0.008 | 0.015801 | −0.48 | 0.633 |
| High temp. control | 0.023 | 0.015892 | 1.46 | 0.153 |

**Table 2. Generalized linear model output of freeze-thaw treatments on cache weight loss to examine the '*frequency hypothesis*' and '*continuous thaw hypothesis*'.** Treatment was used as the predictor variable and proportional weight loss was used as the dependent variable with a quasibinomial distribution.

| Fixed effect | β Estimate | Std. error | t-value | *p*-value |
|---|---|---|---|---|
| Intercept (low freq. freeze-thaw) | 0.918 | 0.007 | 129.48 | < 0.0001 |
| Med freq. freeze-thaw | −0.054 | 0.009 | −6.13 | <0.0001 |
| High freq. freeze-thaw | −0.071 | 0.009 | −8.13 | < 0.001 |
| Control | 0.002 | 0.009 | 0.23 | 0.819 |

frequency freeze-thaw treatment ($z$(inf) = 2.009, $p$ = 0.2; Fig 2B). Caches in the low frequency freeze-thaw treatment did not differ significantly in weight loss from the control, which had the same average temperature ($z$(inf) = −0.23, $p$ = 1.0; Fig 2B).

### Experiment 3: Thaw intensity (*H4*) & freeze intensity (*H5*) hypotheses

Contrary to predictions from *H4*, caches lost significant more weight in the mild thaw treatment compared to the average freeze-thaw treatment but, consistent with predictions from *H5*, caches lost significantly more weight the mild freeze treatment compared to the average freeze-thaw treatment (Fig 2C, Table S8). There was no significant difference in weight loss between caches in the mild thaw treatment and mild freeze treatment ($z$(inf) = 0.2, $p$ = 1.0; Table 3). Caches in the high temperature control lost significantly more weight than caches in the low temperature control ($z$(inf) = −11.0; $p$ < 0.0001; Table 3).

## Discussion

Our experiments provide evidence that, at parameters representative of conditions experienced by Canada jays in Algonquin Provincial Park, Ontario, Canada, freeze-thaw events accelerate the spoilage of caches, as approximated by weight loss over time. Specifically, we found support for the '*exacerbation hypothesis*', '*continuous thaw hypothesis*', and '*freeze intensity hypothesis*' but, as results from experiments 2 and 3 demonstrated, not the '*frequency hypothesis*' or '*thaw intensity hypothesis*'. The lack of support for the '*frequency hypothesis*'

**Table 3. Generalized linear model output of freeze-thaw treatments on cache weight loss to examine the '*thaw intensity hypothesis*' and '*freeze intensity hypothesis*'.** Treatment was used as the predictor variable and proportional weight loss was used as the dependent variable with a quasibinomial distribution.

| Fixed effect | β Estimate | Std. Error | t-value | *p*-value |
|---|---|---|---|---|
| Intercept (ave. freeze-thaw) | 0.933 | 0.003 | 274.13 | < 0.0001 |
| Mild freeze | 0.059 | 0.004 | 13.8 | < 0.0001 |
| Mild thaw | 0.060 | 0.004 | 14.01 | < 0.0001 |
| Low temp. control | 0.068 | 0.004 | 15.92 | < 0.0001 |
| High temp. control | 0.117 | 0.004 | 26.91 | < 0.0001 |

could be because the positive effect of a shorter individual thaw was greater than the negative effect of an additional seven freeze-thaw events (i.e., the value that the two treatments differed by). Proportional weight loss was positively related to the duration of a continuous thaw, corroborating findings in food science literature [34]. Unexpectedly, increasing the maximum temperature of the thaw (i.e., increasing intensity) reduced, not increased, weight loss relative to the other treatments, providing no support for the '*thaw intensity hypothesis*'. The mixed support of how maximum temperatures and frequencies of freeze-thaws influenced spoilage suggest that there are other factors that affect cache quality, such as the intrinsic properties of the meat or microbial communities [62,63]. We discuss these possibilities below.

From the results from experiment 1, which showed that earlier freeze-thaw events resulted in greater weight loss of caches, we can conclude that, for the same above-freezing temperature, spoilage rates are higher post-freeze thaw than pre-freeze. We speculate that the higher rate of spoilage could be attributed to physical destruction that results from the freeze-thaw event, which exacerbates microbial spoilage and autooxidation [4]. Based on studies in food science examining the reproductive nature of bacteria [64,65], we suspect that microbial proliferation was the predominant contributor to the spoilage documented in our experiments. Our results align with previous research but are unique because the month-long durations used in our experiments extended past the typical 10-day durations used in prior food science studies on uncured meats [21,66]. The early freeze-thaw treatment we employed was designed to represent early cold spells that, based on weather data, have occurred in Algonquin Park. These early-season freeze-thaws tend to be followed by longer-than-average thaw periods.

Contrary to previous studies that provided evidence that the number of freeze-thaw events had a negative effect on Canada jay reproductive success [27] and population growth rate [26], we found no support for the '*frequency hypothesis*', where cache spoilage was hypothesized to be positively related to the number of freeze-thaw events. Instead, our experimental results provide support for the '*continuous thaw hypothesis*', suggesting that the duration of individual thaws has more of an impact on cache spoilage than the number of freeze-thaw events. Freezing events destroy the cell membranes, giving microbes access to nutrients in the following thaw. However, the low frequency treatment should have killed fewer bacteria because there were fewer freezing events that would lyse bacterial cells, leaving a larger population of microbes at the start of each thaw [36]. The individual thaw periods were the longest in the low frequency treatment, which would have given more time for the bacteria to proliferate, resulting in greater spoilage compared to the medium and high frequency freeze-thaw treatments. Sutton et al. [27] may have come to the conclusion that the number of freeze-thaw events had a stronger negative effect on Algonquin Park Canada jay population dynamics than mean temperature because the number of freeze-thaw events is strongly correlated with both the duration of individual freeze and thaw phases whereas mean temperature correlates with several climate metrics that may associate in different ways with cache spoilage (Fig S1).

The lack of difference in weight loss between the medium frequency freeze-thaw treatment and the high frequency freeze-thaw treatment may be because a significant proportion of the bacteria were killed after fifteen freeze-thaw events or that there was insufficient time for increased bacterial activity of the surviving population to contribute significantly to spoilage when individual thaw phases were 16 versus 23.5 hours. When temperatures enter hospitable conditions for microbial growth, there is a time lag before bacterial metabolism speeds up [67]. Depending on the bacterial species, food type, and temperature, bacteria have different lag phase durations (LPD) and changes in abundance [4]. Kataoka et al. [67] found that at thawing conditions of 4 °C, the LPD for *Listeria monocytogenes*, a gram-positive bacteria responsible for listeriosis that can lead to hospitalization for humans, was 24 – 48 hrs depending on food type, with bacteria on crabmeat and shrimp having a longer LPD than those on green peas and corn. Another plausible explanation could be that the rate of lipid peroxidation changed over time. Along with several other studies [36–38], Soyer et al. [39] found that peroxidation rates were not linear, with storage duration and temperature having significant effects on the extent of oxidation. These results are significant given that winter minimum temperatures in Algonquin have shifted 8 °C higher, with less intense freezes and longer thaws [33,68]. This shift could have strong negative consequences for cachers of perishable food that rely on persistent subfreezing conditions to maintain cache quality.

We found that, compared to a freeze that reached −6.7 °C, a milder freeze to −4.3 °C resulted in less proportional weight loss, providing support for the '*freeze intensity hypothesis*'. A milder freeze may have led to increased spoilage because freezing rate determines the size of ice crystals formed, which is a critical determinant of food quality [69,70]. The more rapidly temperatures drop below freezing, the smaller the ice crystals that are formed because there is less time for water molecules to migrate extracellularly to the cell membrane before they crystallize [71]. When a food item thaws and ice crystals melt, this leaves smaller holes in cell membranes and consequently less drip loss and more restricted access to nutrients by microbes [5,58,63,71]. Most bacterial growth is inhibited at temperatures of −5 °C or lower, suggesting that the mild freeze treatment did not reach low enough temperatures to fully arrest growth [72]. In contrast, each freeze in the average freeze-thaw treatment likely killed a larger proportion of microbes, reducing the population size that would have resumed activity in the following thaw [48]. The increased average winter temperatures in Canada's boreal ecozones are trending towards milder freezes that are less effective at halting cache spoilage [68].

Unexpectedly, the cold temperature controls that lacked freeze-thaw events lost more weight than all treatment groups except the early freeze-thaw treatment in experiment 1, which had a 211-hour post-freeze thaw. Sublimation from ice to water vapour occurs at temperatures below freezing as unbound water molecules are still available for chemical reactions above −40 °C [5]. Despite the stable, subfreezing environment of the low temperature control, the vapour pressure of the food is greater than in the surrounding air and sublimation can occur to equilibrate the pressure, causing water loss in the cache [73]. In addition, the common aerobic meat microbe, *Pseudomonas*, can become cold-adapted through increased production of exozymes to elevate metabolic activity [74]. The stable and relatively warm conditions of the cold temperature treatment may have also enabled bacterial activity to continue in addition to constant sublimation, resulting in greater proportional weight loss of cold temperature controls relative to freeze-thaw treatment replicates.

A second unexpected result was evidence that refuted the '*thaw intensity hypothesis*'. Caches in the mild thaw treatment lost significantly less weight than those in the average freeze-thaw treatment, despite the maximum temperature being 3.2 °C lower than the average freeze-thaw treatment. Food science literature has shown inconsistent effects of thawing methods on food weight, with some rapid thaw treatments resulting in less drip loss [34] while others increase

drip loss [35]. The treatments here had identical thaw durations, so it is unlikely that the lower proportional weight loss was because the average freeze-thaw treatment was less facilitative of spoilage given its higher maximum temperature [41]. Instead, the more rapidly increasing temperature in the average freeze-thaw treatment may have caused condensation to form on the surface of the chicken possibly because of accelerated denaturation of myofibrillar proteins [75,76]. Partial denaturation and aggregation of myosin can create a 3-dimensional structure that traps water at the surface [77]. In addition, water molecules are naturally attracted to one another because hydrogen bonds confer a cohesive property. Increased water retention due to structural changes of the food itself and the cohesive property of water may have resulted in myosin gelation that facilitates incomplete evaporation of water at the surface of the cache [76,78]. The gel formation concentrates protein at the surface of the cache, which could make it readily available for microbes to use in thawing conditions. In addition, depending on the species, the palatability of the cache may decrease as a result of this gel formation.

Our series of experiments suggest that the timing, but not the frequency, of freeze-thaw events and the duration of individual thaw periods post-freeze are the strongest contributors to hoard rot for high-latitude food cachers. Post-thaw spoilage rates following a freeze are higher, but it appears that the spoilage rates only increase after a minimum duration of above-freezing temperatures has been maintained, likely allowing autooxidation and microbial rates to increase to significant rates. Our results contribute towards understanding the impact warming conditions may have on species that cache perishable food, especially those that rely on stored food for over-winter survival. The negative effects of warming fall conditions [27,28] could nevertheless be mitigated by earlier frost-free periods in the spring [68] if there is a temporal advancement of emergent spring resources that provide a valuable alternative food source to highly degraded caches. Canada jays in Algonquin Park [79] and Denali National Park, Alaska (63°44' N, 148°54' W) have been observed to divert their foraging away from arboreal caches to emergent resources on the ground as soon as snow cover disappears [80]. The future distribution and success of perishable food cachers will depend on how rising temperatures affect the quality of critical perishable food stores, the flexibility of food-caching species to switch food sources, and the availability of suitable habitat at higher latitudes.

## Supporting information

**S1 Table. The number of freeze-thaw events in Oct., Nov. and Dec. from 2004 – 2019.** Historical climate data were obtained from Environment Canada at the Algonquin Provincial Park East Gate weather station (45°32'N, 78°54'W; https://weather.gc.ca/city/pages/on-29_metric_e.html).
(PDF)

**S2 Table. The summary of freeze and thaw phases in the month of Nov. from Algonquin Provincial Park East Gate weather station (45°32'N, 78°54'W; 2004 – 2019).** The number indicates the number of individual freeze-thaw events. Duration indicates the average duration (hrs) of a freeze or thaw in a given year. The average minimum and maximum temperatures provide the average lowest temperature reached in freezes and highest temperature in thaw events in a given year.
(PDF)

**S3 Table. Hourly programming schedule for the four freezers used in experiment 1 to test the predictions of the 'exacerbation hypothesis'.** Each freezer was dedicated to simulating either the early freeze-thaw treatment, late-freeze-thaw treatment, low temperature control (held at −4.9 °C, no freeze-thaw events), or high temperature control (held at 1.1 °C,

no freeze-thaw events). All simulations were 720 hours long and both treatments had eight freeze-thaw events each.
(PDF)

**S4 Table. Hourly programming schedule for all freezers used in experiment 2 to test the predictions of the '*frequency hypothesis*' and the '*continuous thaw hypothesis*'.** Each freezer was dedicated to simulating either the low frequency freeze-thaw treatment (8 freeze-thaw events), medium frequency freeze-thaw treatment (15 freeze-thaw events), high frequency freeze-thaw treatment (22 freeze-thaw events), or the control (held at −1.6 °C, no freeze-thaw events). All simulations were 720 hours long and all treatment groups had an identical total duration below freezing (368 hrs) and above-freezing (352 hrs) post-freeze.
(PDF)

**S5 Table. Hourly programming schedule for all freezers used in experiment 3 to test the predictions of the '*thaw intensity hypothesis*' and '*freeze intensity hypothesis*'.** Each freezer was dedicated to simulating either the average freeze-thaw treatment (freezing and thawing rate of 0.8 °C/hr), mild freeze treatment (freezing rate of 0.4 °C/hr; thawing rate of 0.8 °C/hr), mild thaw treatment (freezing rate of 0.8 °C/hr; thawing rate of 0.4 °C/hr), low temperature control (held at −4.9 °C, no freeze-thaw events), or high temperature control (held at 1.1 °C, no freeze-thaw events). All simulations were 720 hours long and each treatment group had eight freeze-thaw events.
(PDF)

**S6 Table. Weight loss of caches in experiment 1 that tested the predictions of the '*exacerbation hypothesis*'.** Caches consisted of 1.20g of raw chicken breast placed between two pieces of black spruce (*Picea mariana*) bark.
(PDF)

**S7 Table. Weight loss of caches in experiment 2 that tested the predictions of the '*frequency hypothesis*' and the '*continuous thaw hypothesis*'.** Caches consisted of 1.20g of raw chicken breast placed between two pieces of black spruce (*Picea mariana*) bark.
(PDF)

**S8 Table. Weight loss of caches in experiment 3 that tested the predictions of the '*freeze intensity hypothesis*' and '*thaw intensity hypothesis*'.** Caches consisted of 1.20g of raw chicken breast placed between two pieces of black spruce (*Picea mariana*) bark.
(PDF)

**S1 Fig. Correlation coefficients between November climate variables calculated based on the angular order of eigenvectors.** * indicates $p < 0.05$, ** indicates $p < 0.01$, and *** indicates $p < 0.001$. 'Total freeze duration (hrs)' is total hours below freezing point for a given year, 'number of freeze-thaws' is total number of freeze-thaw events that occurred in Nov. for a given year, 'mean min. temp. (freeze)' is mean of the lowest temperatures reached across individual freeze phases for a given year, 'mean hourly temp.' is mean temperature of the 720 hrs for each year, 'total thaw duration (hrs)' is total hours above freezing point for a given year following the first freeze, 'mean max. temp. (thaw)' is mean of the maximum temperatures reach across individual thaw phases for a given year, 'mean thaw duration (hrs)' is mean duration in hours for an individual thaw for a given year, and 'mean freeze duration (hrs)' is mean duration in hours of individual freeze phases for a given year. Calculations were based on hourly climate data for the month of Nov. from Algonquin Provincial Park East Gate (45°32'N, 78°54'W; 2004 − 2019).
(TIF)

## Acknowledgements

We thank Dan Strickland for his important contributions to the development of this manuscript, Sue Couling from the University of Guelph Crop Science Department for allowing us access to, and for aiding with, programmable freezers, and the Guelph Arboretum for allowing us to sample bark from black spruce trees.

## Author contributions

**Conceptualization:** Karen Ong, D. Ryan Norris.

**Data curation:** Karen Ong.

**Formal analysis:** Karen Ong.

**Funding acquisition:** D. Ryan Norris.

**Methodology:** Karen Ong, D. Ryan Norris.

**Resources:** Karen Ong, D. Ryan Norris.

**Supervision:** D. Ryan Norris.

**Validation:** Karen Ong.

**Visualization:** Karen Ong.

**Writing – original draft:** Karen Ong.

**Writing – review & editing:** D. Ryan Norris.

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
