## [Decision Letter · Decision Letter 0]

25 Nov 2024

PONE-D-24-39177Experimental evidence demonstrating how freeze-thaw patterns affect spoilage of perishable cached foodPLOS ONE

Dear Dr. Karen Ong,

Thank you for submitting your manuscript to PLOS ONE. After careful consideration, we feel that it has merit but does not fully meet PLOS ONE’s publication criteria as it currently stands. Therefore, we invite you to submit a revised version of the manuscript that addresses the points raised during the review process.

**ACADEMIC EDITOR: Please reply for all reviewers questions and follow all their suggestions. **

We look forward to receiving your revised manuscript.

Kind regards,

Ayman A. Swelum

Academic Editor

PLOS ONE

Journal Requirements:

2. Thank you for stating the following in your Competing Interests section: [The authors have declared that there are no competing interests.]. Please complete your Competing Interests on the online submission form to state any Competing Interests. If you have no competing interests, please state "The authors have declared that no competing interests exist.", as detailed online in our guide for authors at http://journals.plos.org/plosone/s/submit-now This information should be included in your cover letter; we will change the online submission form on your behalf.

3. We note that your Data Availability Statement is currently as follows: [All data are contained within the manuscript and supporting information files.] Please confirm at this time whether or not your submission contains all raw data required to replicate the results of your study. Authors must share the “minimal data set” for their submission. PLOS defines the minimal data set to consist of the data required to replicate all study findings reported in the article, as well as related metadata and methods (https://journals.plos.org/plosone/s/data-availability#loc-minimal-data-set-definition). For example, authors should submit the following data: - The values behind the means, standard deviations and other measures reported; - The values used to build graphs; - The points extracted from images for analysis. Authors do not need to submit their entire data set if only a portion of the data was used in the reported study. If your submission does not contain these data, please either upload them as Supporting Information files or deposit them to a stable, public repository and provide us with the relevant URLs, DOIs, or accession numbers. For a list of recommended repositories, please see https://journals.plos.org/plosone/s/recommended-repositories. If there are ethical or legal restrictions on sharing a de-identified data set, please explain them in detail (e.g., data contain potentially sensitive information, data are owned by a third-party organization, etc.) and who has imposed them (e.g., an ethics committee). Please also provide contact information for a data access committee, ethics committee, or other institutional body to which data requests may be sent. If data are owned by a third party, please indicate how others may request data access.

5. We notice that your supplementary [Figure S1] are included in the manuscript file. Please remove them and upload them with the file type 'Supporting Information'. Please ensure that each Supporting Information file has a legend listed in the manuscript after the references list.

Reviewers' comments:

Reviewer's Responses to Questions

**Comments to the Author**

1. Is the manuscript technically sound, and do the data support the conclusions?

Reviewer #1: Partly

Reviewer #2: Yes

2. Has the statistical analysis been performed appropriately and rigorously? 

Reviewer #1: Yes

Reviewer #2: Yes

3. Have the authors made all data underlying the findings in their manuscript fully available?

Reviewer #1: No

Reviewer #2: Yes

4. Is the manuscript presented in an intelligible fashion and written in standard English?

Reviewer #1: Yes

Reviewer #2: Yes

5. Review Comments to the Author

Reviewer #1: 1. Abstract has to be written in a manner that encompass why, how and what obtained from the current research.

2. Methods has to be written clearly to be replicable

3. The findings have to be presented up to the standard.

4. Discussion is good but need improvement.

5. Conclusion must be based on the findings of the current study.

Reviewer #2: 1. The title is ok and related to the manuscript.

2. Abstract is quite good. Maybe can include some method.

3. Introduction is ok.

4. Methods are well explained.

5. Results are ok. The graphs and diagrams can be view publicly.

6. The discussion is quite good.

6. PLOS authors have the option to publish the peer review history of their article (what does this mean?). If published, this will include your full peer review and any attached files.

Reviewer #1: No

Reviewer #2: No

---

## [Author Response · Author response to Decision Letter 0]

6 Jan 2025

Thank you for these comments. Please see below for our responses (text following "Author response:") to each of the comments provided by the two reviewers. We think the manuscript has been improved considerably.

Thank you,

Karen Ong, Ryan Norris

Reviewer #1:

1. Abstract has to be written in a manner that encompass why, how and what obtained from the current research.

Author response: We have extensively revised the abstract to make sure all of these elements are properly communicated. Specifically, we have added more information on the rationale for the experiment, as well as additional methodological details.

2. Methods has to be written clearly to be replicable

Author response: We have also made extensive revisions the methods to improve clarity. We have also added tables in the supplementary material that show hourly programming schedules for all experiments.

3. The findings have to be presented up to the standard.

Author response: Our apologies. We are not sure what the reviewer means here. We have edited the results to improve clarity, but the presentation of the results includes all necessarily statistical moments and output from the models.

4. Discussion is good but need improvement.

Author response: Given the reviewer has not been specific here, we have carefully gone through the discussion to improve clarity in several places. As a result, we think the discussion has improved considerably.

5. Conclusion must be based on the findings of the current study.

Author response: All our conclusions are based on the results of the experiments presented in this paper.

Reviewer #2:

1. The title is ok and related to the manuscript.

Author response: Thank you.

2. Abstract is quite good. Maybe can include some method.

Author response: Thank you. As suggested, we have included additional information in the abstract about the methods used in our experiments.

3. Introduction is ok.

Author response: Thank you.

4. Methods are well explained.

Author response: Thank you.

5. Results are ok. The graphs and diagrams can be view publicly.

Author response: Thank you.

6. The discussion is quite good.

Author response: Thank you. We appreciate the positive comments.

---

## [Decision Letter · Decision Letter 1]

26 Jan 2025

Experimental evidence demonstrating how freeze-thaw patterns affect spoilage of perishable cached food

PONE-D-24-39177R1

Dear Dr. Ong,

We’re pleased to inform you that your manuscript has been judged scientifically suitable for publication and will be formally accepted for publication once it meets all outstanding technical requirements.

Kind regards,

Ayman A Swelum

Academic Editor

PLOS ONE

Additional Editor Comments (optional):

Reviewers' comments:

Reviewer's Responses to Questions

**Comments to the Author**

1. If the authors have adequately addressed your comments raised in a previous round of review and you feel that this manuscript is now acceptable for publication, you may indicate that here to bypass the “Comments to the Author” section, enter your conflict of interest statement in the “Confidential to Editor” section, and submit your "Accept" recommendation.

Reviewer #1: All comments have been addressed

2. Is the manuscript technically sound, and do the data support the conclusions?

Reviewer #1: Yes

3. Has the statistical analysis been performed appropriately and rigorously? 

Reviewer #1: Yes

4. Have the authors made all data underlying the findings in their manuscript fully available?

Reviewer #1: Yes

5. Is the manuscript presented in an intelligible fashion and written in standard English?

Reviewer #1: No

6. Review Comments to the Author

Reviewer #1: (No Response)

7. PLOS authors have the option to publish the peer review history of their article (what does this mean?). If published, this will include your full peer review and any attached files.

Reviewer #1: No

---

## [Editor Report · Acceptance letter]

PONE-D-24-39177R1

PLOS ONE

Dear Dr. Ong,

I'm pleased to inform you that your manuscript has been deemed suitable for publication in PLOS ONE. Congratulations! Your manuscript is now being handed over to our production team.

Kind regards,

on behalf of

Professor Ayman A Swelum

Academic Editor

PLOS ONE